# Formulation of Heat-Induced Whey Protein Gels for Extrusion-Based 3D Printing

**DOI:** 10.3390/foods10010008

**Published:** 2020-12-22

**Authors:** Valeska F. Sager, Merete B. Munk, Mikka Stenholdt Hansen, Wender L. P. Bredie, Lilia Ahrné

**Affiliations:** 1Department of Food Science, Faculty of Science, University of Copenhagen, 10 DK-1165 Frederiksberg C, Denmark; vfs@food.ku.dk (V.F.S.); bogelund@food.ku.dk (M.B.M.); wb@food.ku.dk (W.L.P.B.); 2Innovation Centre, Arla Foods, Skejby, 8260 Viby, Denmark; mstha@arlafoods.com

**Keywords:** extrusion-based 3D printing, emulsion gels, microparticulated whey protein, micellar casein isolate

## Abstract

This study investigated the extrusion-based 3D printability of heat-induced whey protein gels as protein rich food inks. In particular, the effects of ionic strength by the addition of NaCl (0–250 mM), protein content (10%, 15%, 20%), fat content (0%, 10%), and partial substitution of whey protein isolate (WPI) with microparticulated whey protein (MWP) or micellar casein isolate (MCI) on printability were assessed. Texture analysis, specifically Young’s modulus, rheological measurements including yield stress, and creep–recovery behavior were used to characterize the gels. Modifications of the formulation in terms of ionic strength, increased protein content, and the formation of emulsion gels were insufficient to maintain a continuous extrusion process or shape stability after printing. However, the substitution of WPI with MWP created more viscoeleastic gels with improved printability and shape retention of the 3D cube structure after deposition. The partial replacement of WPI with MCI led to phase separation and 3D-printed cubes that collapsed after deposition. A narrow range of rheological material properties make WPI and MWP emulsion gels promising food inks for extrusion-based 3D printing.

## 1. Introduction

The growing population of elderly and other consumer groups with special dietary needs, such as hospitalized patients, demands the development of versatile, sensory attractive, energy-dense, and protein-rich nutritional products. In this context, three-dimensional (3D) printing of foods is a technology with substantial potential. Three-dimensional printing allows the production of small batches of food products with personalized nutritional composition and sensory preferences. Fresh, locally produced, and customized 3D-printed food products have shown to enhance the food uptake by patients with reduced appetite [1].

Three-dimensional (3D) food printing can be performed by binder jetting, inkjet printing, selective laser sintering, and extrusion-based processes. These technologies have different feedstock requirements and unique areas of application as comprehensively reviewed by Liu et al. [1]. The extrusion-based technology is the most commonly applied technology to 3D print semi-solid materials [2]. In extrusion-based 3D printing, the food material, also called food ink, is extruded out of a narrow nozzle. To ensure that this process does not destroy the material structure, the food ink must have suitable mechanical and rheological properties including an appropriate strength, viscoelasticity, and yield stress [3]. So far, a wide range of foods have been 3D printed using the extrusion-based technology and evaluated based on shape stability after printing. This includes surimi gel [4], vegemite [5], lemon juice gel [6], pavlova [7], smoothie paste [8], mushrooms [9], mashed potato, marzipan and bean paste [10], different fiber-rich materials [11], and pectin-based gels [12]. A number of studies have also focused on 3D printing of dairy-based foods inks, such as pastes made from milk and whey protein ingredients [2,11,13,14] and processed cheeses [15].

Whey proteins have the ability to form gels, allowing the design of foods with attractive sensory properties, bioavailability [16], and the formation of complex shapes by 3D printing [17]. To our knowledge, this is the first study to focus on formulating whey protein gels for extrusion-based 3D printing. Several strategies could be explored to design gels for 3D printing, including the development of materials that gelled after extrusion. In this study, we want to explore the possibility to modify the gel rheological properties of the ink by changing the formulation, in order to make possible the extrusion of a gel through a narrow nozzle of a 3D printer to produce a soft gel after printing.

Whey protein isolate (WPI) is a dairy ingredient with a high protein content that depending on concentration, ionic strength, and pH can form strong gels during heat treatment, due to electrostatic, hydrophobic, and thiol-disulfide exchange reactions between denatured whey proteins [18,19]. Three-dimensional (3D) printed gels require a self-supported matrix after deposition, and therefore, a suitable WPI gel-based ink will need to resist the shear forces during extrusion without fracturing into pieces. This means that the ink gel structure needs to be able to reform into a stable network immediately after being forced through the narrow nozzle. Strategies to modulate the structure and rheological properties of whey protein gels that have been previously reported but not explored in the context of 3D printing include modifications of ionic strength [20] or the addition of fat to create an emulsion gel [21]. Furthermore, previous studies have shown that microparticulated whey protein (MWP) or micellar casein isolate (MCI) are high protein ingredients (≈80% protein) used in the formulation of a variety of foods and drinks, that contrary to WPI do not form strong gels [22,23], making them suitable ingredients to modulate WPI-based gel formulations for 3D printing.

To pave the way for developing high-protein food inks, the aim of this study was to explore the extrusion-based 3D printability of heat-induced high protein WPI gels. This was done by modifying their formulation, in terms of changing the ionic strength, increasing WPI content, adding fat to form an emulsion gel, or substituting part of WPI with other protein ingredients that modify the rheological properties of the gel, such as MWP and MCI. The textural and rheological properties of the gels were evaluated by compression tests, oscillatory stress sweeps, and modeling of creep recovery behavior. The focus was on performance of the gel-based inks, and therefore, printability was only assessed visually.

## 2. Materials and Methods

### 2.1. Materials

Whey protein isolate (Lacprodan^®^ DI-9224) (WPI, protein content 89.5%), microparticulated whey protein (MWP, protein content 81.0%), and micellar casein isolate (MCI-85) (MCI, protein content 83.3%) were provided by Arla Foods Ingredients (Viby J, Denmark). The chemical composition and physicochemical characteristics of these proteins have been described in detail by Liu et al. [24]. Whipping cream (38% fat) (Arla Foods amba, Slagelse Dairy, Denmark) was purchased from a local supermarket.

### 2.2. Experimental Design

The experimental work consisted of three parts. The first part investigated the effect of ionic strength on gel stiffness and 3D printability of WPI gels (10% protein). The second part studied the effect of protein content (10, 15, and 20% protein from WPI) and fat (0% and 10%) on the printability and gel stiffness of WPI gels (0% fat) and WPI emulsion gels (10% fat). In the third part, the possibility to modify the rheological properties and printability of emulsion gels composed by 10% fat and 10% protein was investigated by substituting parts of WPI with microparticulated whey protein (MWP) and micellar casein isolate (MCI). Table 1 describes details of the experimental design factors and selected levels used in the experimental part three. At least two independent experiments were performed.

### 2.3. Gel Preparation

Based on the protein content in WPI, MWP, and MCI protein powders, stock solutions of 30%, 20%, and 10% protein (*w/w*), respectively, were mixed with Mili-Q water and left overnight at room temperature (25 °C) to allow the hydration of proteins. Subsequently, stock solutions were diluted with Mili-Q water and mixed to obtain the desired protein concentration and composition. For emulsion gels, whipping cream was added to reach a fat content of 10% (*w/w*). WPI gels (10% *w/w* protein) with varying ionic strength were prepared from stock solutions of 0, 50, 150, and 250 mM NaCl. Before exposing all samples to severe mixing at 13,500 rpm for 30 s using an Ultra Turrax, the pH was adjusted to 7.0 with 0.1 M HCl or 0.1 M NaOH. Conductivity was measured with a conductivity meter (MeterLab ION450) and was found to be 2.7, 6.5, 13.7, and 22.9 mS cm^−1^ in gels with 0, 50, 150, and 250 mM NaCl, respectively. The mixed samples were filled into pre-lubricated (rapeseed oil) 30 mL cartridges with a diameter of 22 mm. The cartridges were subsequently heated at 80 °C for 30 min in a water bath to induce a full developed gel network [25]. The gels were cooled and stored overnight at room temperature before printing and further analyses.

### 2.4. 3D Printing

An extrusion-based 3D printer (Focus 3D Food Printer, byFlow, Eindhoven, The Netherlands) was used to assess printability of the gels. Printing was carried out at room temperature using an extrusion nozzle of 1.6 mm in diameter. A cube predesigned by byFlow with dimensions of 20 × 20 × 10 mm (length × width × height) was printed layer by layer, with a total of 10 layers each consisting of 14 lines. The printing speed was pre-selected based on the printing requirements for the nozzle. The performance of the food inks was assessed during the printing process and reported by pictures. No quantification of the printability of the 3D printed gels was done, as the focus of this study was the performance of the gel based inks during 3D printing.

### 2.5. Gel Stiffness

The gel stiffness, expressed by Young’s modulus, E (Pa), was measured before and after 3D printing using a TA.XTplus Texture Analyzer (Stable Micro Systems, Surrey, UK) fitted with a non-lubricated cylindrical plate (diameter 75 mm) and equipped with a 50 kg load cell. Each sample was compressed to 90% of its initial height at a constant deformation rate of 1 mm s^−1^. Young’s modulus was determined by the slope of true stress (*σ*) at true strain (*ε*) for compressions up to 5% using following equations:(1)$$\sigma \left(t\right)=\frac{F\left(t\right)}{A_{i}}\cdot \frac{H\left(t\right)}{H_{i}}$$
(2)$$\varepsilon \left(t\right)=-ln\frac{H\left(t\right)}{H_{i}}$$
where *F(t)* is the force at a given time, *A_i_* is the cross-sectional area of the sample, *H(t)* is the height of the sample at time *t*, and *H_i_* is the initial sample height. This method was adapted from Çakir et al. [26].

Prior to compression measurements, the unprinted gels were removed from the cartridges and cut into cylindrical pieces (22 mm diameter and 20 mm height) using a steel wire. For printed gels, the compression measurements were carried out immediately after printing. Each sample was measured in triplicate, and the results are presented as the mean ± std. deviation.

### 2.6. Rheological Analysis

The rheological behavior of gels was characterized using a DHR-2 rheometer (TA Instruments, New Castle, DE, USA) at 20 °C equipped with serrated parallel plates with a diameter of 25 mm. Cylindrical gels with a diameter of 22 mm were cut in slices with a height of 10 mm using a steel wire. After sample loading, the sample was allowed to rest for 30 s before measurement.

### 2.7. Yield Stress

To determine the yield stress of unprinted gels, oscillation stress sweeps were performed with logarithmically increasing shear stress from 0.1 to 1000 Pa at a fixed frequency of 1 Hz. The cross-over point where elastic modulus (G’), equaled the viscous modulus (G’’) was taken as the yield stress [3,27]. Each sample was measured in triplicate.

### 2.8. Creep–Recovery

Measurements were conducted on gels before printing by applying an instantaneous shear stress on the sample for 180 s (creep time) whereupon the stress was removed and sample recovery was monitored for additional 180 s (recovery time). Two cycles of creep–recovery test were performed in the linear viscoelastic region of 1 Pa, which was previously determined during oscillation stress sweeps. Each sample was measured in duplicate. The recovery strain was determined as:RS (%) = (final strain − initial strain)/initial strain * 100.(3)

Creep–recovery data plotted as strain (%) versus time (min) were fitted to the Simplex Nelder–Mead algorithm implemented in Matlab. This method has been earlier defined and used by Spotti et al. [28] and allows estimation of the parameters α, λ_1_, and λ_2_, which characterize the viscoelasticity of a material. α describes the derivative order, which ranges between 0 and 1. A = 1 is characteristic of a purely viscous material, whereas α = 0 presents a purely elastic material. The creep constant λ_1_ describes the first compression, whereas λ_2_ is an index for the recovery after the compression of a material. The difference between λ_1_ and λ_2_ is used to describe the recovery of the sample, where a large value indicates non-recoverable deformation of the sample, in other words structure damage. In order to determine α, λ_1_, λ_2_, the best fit, with the smallest root mean squared error (RMSE) of 100 runs was chosen.

### 2.9. Statistical Analysis

The statistical software JMP 13 (SAS, Cary, NC, USA) was used to analyze data, which are presented as the mean including standard deviations.

## 3. Results and Discussion

### 3.1. Effect of Ionic Strength on Printability of WPI Gels

In the first part of this work, we assessed whether changes in ionic strength by increasing concentration of NaCl improved the printability of WPI gels. The differences in gel stiffness before and after 3D printing described by Young’s modulus are shown in Figure 1. At low ionic strength and far from the isoelectric point (≈pH 5.2), WPI solutions are stabilized by electrostatic repulsion between globular proteins such as β-lactoglobulin [29]. The heating of WPI solutions (10% protein) at pH 7, without NaCl addition, formed transparent gels, which were earlier described as fine stranded [29,30]. These gels had a low gel stiffness (Young’s modulus: 1783 ± 252 Pa), and although they could be printed, the 3D-printed cube showed non-distinguishable layers and structure collapse after printing. Increasing the ionic strength by the addition of NaCl as previously reported increased the stiffness of the gels; however, the printability was not significantly improved. Gels containing 50 mM NaCl were harder, and they could be extruded in a continuous stream through the nozzle but did not form cubes with well-defined edges. Gels containing 150 mM NaCl became too hard and fractured during printing (Figure 1).

Increasing ionic strength produced gels with increased stiffness up to a certain point corresponding to 150 mM NaCl, whereupon the stiffness drastically decreased, as previously reported [20,31]. Up to around 150 mM NaCl, salt ions appeared to reduce the electrostatic repulsion of WPI by neutralizing some of the charges on the surface of the proteins and thereby enhancing protein–protein interactions and consequently gel stiffness [32]. However, higher ionic strength (250 mM NaCl) might retard protein unfolding and create weaker gels. These gels were found to be opaque with a coarse and brittle structure, which has been earlier found to be a result of significantly larger protein aggregates than seen in fine-stranded gels [31]. Gels containing 250 mM NaCl had lower gel stiffness (Young’s modulus: 5909 ± 638 Pa) than the other gels; however, they still broke into smaller fragments during extrusion, showing extensive syneresis and could not form a well-defined cube structure. From a 3D printing perspective, it is important to take into account that gel stiffness is dependent on two opposing phenomena. On the one hand, the increase in gel strand thickness and subsequently gel stiffness is caused by the reduction in electrostatic repulsion and the formation of intramolecular disulfide bonds. On the other hand, there is a reduction of available binding sites as the gel strand thickness increases, which eventually can decrease the gel stiffness [31]. The extrusion of the gels through the narrow nozzle of the 3D printer caused a non-recoverable structural deformation of the gels as observed by the significantly reduction in the stiffness of the gels even for gels with low initial stiffness (Figure 1). In contrary to studies with surimi gels [4], increasing the ionic strength was not sufficient to improve the printability of WPI gels. Additionally, the gel stiffness per se was shown not to be a good indicator to predict the printing behavior of these gels.

### 3.2. Effect of WPI Concentration and Addition of Fat on Printability of WPI Emulsion Gels

The second part of this study evaluated how protein concentration and the addition of fat, which creates an emulsion gel, influence gel stiffness and printability. Young’s modulus of WPI gels containing 10, 15, and 20% protein formulated with and without fat can be seen in Figure 2. Gels with 10% WPI were very soft reflected by a very low Young’s modulus (Figure 2a), and the printed cube had no distinct layers and a high degree of syneresis. Increasing the protein content from 10% to 15% and even further to 20% increased the Young modulus to 25 kPa and 75 kPa, respectively, making the gels too stiff to be extruded by 3D printing, due to an increase in intra- and inter-chain disulfide interactions between denatured whey proteins [28,33].

The addition of 10% fat into the WPI gel network formed emulsion gels with a significant higher gel stiffness (25–230 kPa). Due to the presence of fat, the 10% WPI protein gel could be printed, but fragmented gels are deposited during printing (Figure 2b). The 3D-printed gel did not have the expected shape of a well-defined layered object; however, from a food culinary experience, it may be an attractive product. It has previously been shown that introducing fat in a WPC gel formulation increased the gel strength at similar protein levels [21]. As active fillers, the protein layer found on the fat globule surface interacts with the dispersed protein phase, whereas fat globules trapped inside the gel matrix with little interaction with the surrounding matrix act as inactive fillers [21,34,35]. Furthermore, fat globules increased the stiffness in whey protein emulsion gels that have a low initial value [36]. At protein contents of 15 and 20%, the gel matrix is substantially stiffer, and the effect of fat addition on the increase of gel stiffness is less pronounced, which explained the similar gel stiffness of the 15% and 20% WPI protein emulsion gels (Figure 2b). In order to better understand the printing behavior of these gels, as previously described [3,27], the yield stress, representing the minimum force required to initialize sample flow, storage modulus G’, and loss modulus G’’, were determined (Table 2). In all formulations, G’ > G’’ in the linear viscoelastic region (0.1–1 Pa), meaning that the gel strength and elasticity dominated over the non-recoverable or viscous part of the material [37]. Rheological analysis showed that gels with 10% protein from WPI had a low yield stress (≈140 Pa) and low G’ (≈110 Pa), which explains why the extruded layers were not strong enough to support the layers deposited on top. In general, 3D printing requires materials with a low yield stress in order to make the material flow out of the nozzle, but if the yield stress is too low, the material is not self-supportive and collapses by the weight of additional layers [3]. The emulsion gels with a protein content of 15% and 20% showed the highest yield stress (>1000 Pa), gel stiffness (Young’s modulus >200 kPa), and G’ (>40,000 Pa) and therefore were not printable (Table 2) with the printer used in this work. Thus, strategies to soften the gel structure by adding other protein ingredients were explored for these emulsion gels.

### 3.3. Effect of Addition of MWP or MCI on Printability of WPI Emulsion Gels

As emulsion gels showed relevant behavior during 3D printing, in the third part of this study, the printability of WPI emulsion gels with 10% protein and 10% fat was improved by substituting parts of WPI with MWP and MCI while maintaining a total protein content of 10%. Microparticulated whey proteins are spherical particles formed from whey protein concentrate by thermal and mechanical processing. MWP are almost completely denatured and aggregated proteins with an average size of 0.5–10 µm [38]. The total content of native whey in MWP has been found to be between 2.8 and 14.5% [39], and the content of native β-lg in MWP can be as low as 0.3% [22]. The low content of native β-lg and thus fewer active sites for aggregation and binding with other proteins has shown to be advantageous in improving the creaminess and smoothness of various foods. Thus, MWP has found application as a fat replacer, texturizer, and stabilizer in e.g., deserts, sauces, and dressings [40]. 

Results obtained from texture analysis before and after 3D printing of WPI emulsion gels supplemented by MWP can be seen in Figure 3, and the rheological properties can be seen in Table 2. Whereas emulsion gels composed only of WPI had tendencies to break into smaller fragments during extrusion, the addition of 2% protein from MWP lead to the extrusion in a continuous stream. The printed cube had well-defined layers that made up a self-supporting structure that could be moved without breaking. The incorporation of 2% MWP did not lead to a significant decrease in gel stiffness before printing compared to 10% WPI emulsion gels.

This can be attributed to the good water-holding capacities of MWP [41] and small amounts of native whey protein present in MWP, which could help incorporation into the emulsion gel matrix [38]. Furthermore, no syneresis was seen, which has been earlier observed when introducing small amounts of MWP during cheese production [41,42]. Substituting 3.5% WPI with MWP decreased gel stiffness before printing with approximately 70% (from 23,795 ± 2264 Pa to 6825 ± 1072 Pa), which yielded a stable cube structure at deposition with smooth layering (Figure 3). Rheological analysis showed that the partial substitution of WPI by MWP decreased the yield stress and led to a decrease in storage modules G’ and loss modulus G’’ (Table 2). This may be explained by MWP weakening the emulsion gel network, leading to a creamier texture [42], which in our case meant an improved extrudability. Substituting 50% of WPI with MWP led to micro-phase separation with MWP flocking and settling at the bottom of the cartridges. This could be a result of steric hindrances or thermodynamic incompatibilities between β-lg and MWP [43]. These formulations did not gel, and consequently, printing was not possible.

Another possibility to improve the printability of emulsion gels was the substitution of WPI by micellar casein isolate (MCI). Due to its high protein content, nutritional value, and functional properties, MCI is used to fortify foods or added during cheese manufacturing. MCI is further known to be relatively heat stable [44]. Commercial MCI is composed of around 83% protein from intact casein micelles and small amounts of whey proteins, and therefore, gel-strengthening interactions with WPI in heat-induced gels are expected to be limited [24]. Substituting WPI with 2% and 3.5% protein from MCI, while maintaining a total protein content of 10%, drastically decreased gel stiffness before printing compared to 10% WPI emulsion gels (Figure 4). These formulations were more easily extrudable; however, the 3D-printed cube with 3.5% MCI melted after deposition showed syneresis and no distinct layering.

The dissolution behavior of MCI powder is time, concentration, and temperature dependent [45]. Therefore, MCI was hydrated overnight before being mixed with other ingredients and heated to form a gel. Still, MCI was not sufficiently incorporated into the gel network, which could be seen from color differences in the printed emulsion gel. Similar to the gels produced with MWP, gels with >3.5% protein from MCI (Table 1) were not printable, as phase separation hindered gelation. 

Oscillatory stress sweeps revealed that the incorporation of 3.5% MCI into WPI emulsion gels decreased yield stress to <300 Pa and G’ to <750 Pa, which was substantially lower than in 10% WPI emulsion gels: ≈800 Pa and ≈2750 Pa, respectively (Table 2). This declined shape stability, as the gels became too weak to support the layers extruded on top, and subsequently, the cubes collapsed. It can be argued that the reduced concentration of WPI caused the reduction in gel stiffness as less disulfide cross-links are formed [46]. However, this does not explain the apparent differences seen between formulations with MWP and MCI, respectively, as they contained the same concentrations of WPI. Gelling was performed at pH 7, which is close to the native pH of bovine milk of 6.7. Here, caseins are negatively charged and stabilized by electrostatic repulsion, which prevents aggregation into larger polymers [23]. Upon heating, MCI might be incorporated into an emulsion gel network formed by WPI and fat, but it may react more as an inert filler than as an active ingredient. Emulsion gel formulations including MCI were generally softer, inhomogeneous, and less printable compared to formulations including MWP.

The results show that a narrow range of textural and rheological material properties have to be met to facilitate continuous material extrusion and desirable layering in the present 3D printing system. This includes a Young’s modulus before printing between 7 and 22 kPa and a yield stress between 382 and 525 Pa as obtained in MWP(2) and MWP(3.5) emulsion gels. Emulsion gels outside this range were either too soft to maintain a stable structure after deposition or too stiff and ruptured during extrusion or were not extrudable at all. 

### 3.4. Creep–Recovery Behaviour of WPI Emulsion Gels Substituted with MWP and MCI

To better understand the viscoelastic behavior of WPI emulsion gels containing MWP and MCI, a creep–recovery test was performed. The creep–recovery curves (Figure 5) showed that all gels had typical viscoelastic properties as described by Qian and Kawashima [47], but they differed in the maximum strain. The gels containing MWP showed a higher strain than the corresponding gels containing MCI, whereas the lowest strain was observed for emulsion gels formed solely with WPI. The higher the amount of MCI or MWP, the higher the maximum strain obtained, i.e., the structure is characterized by more viscous elements and less rigid elastic elements.

The creep–recovery behavior can be further described by the recovery strain (RS) [28], which decreases as the material loses its ability to regain its shape after stress application (Table 3). Emulsion gels containing 10% WPI showed full recovery (RS 100%) in the first and second creep. In the applied stress range (1 Pa), these gels were shown to be highly elastic. As previously seen, these gels were also harder and ruptured during printing. These results indicated that the combination of hard and highly elastic gels at low stresses was not suitable for 3D printing. In addition to MCI(2), emulsion gels substituted with MWP and MCI recovered less (RS < 93%) and lost part of their elasticity, seen by a decline in RS after the second creep. However, as they were less stiff, they allowed continuous extrusion.

Modeling of creep–recovery data using a fractional derivative modeling approach [28] allowed further describing the viscoelasticity of the gels. The parameters of the model representing the degree of elasticity (α), the inverse of the gel elastic modulus during creep (λ_1_) and recovery (λ_2_) are presented in Table 3. All gels had α values ranging from 0.083 to 0.185, which is characteristic for an elastic material. In gels formulated with MCI or MWP, α increased compared to 10% WPI emulsion gels, which indicated the formation of less elastic gels. The difference between the creep and recovery constant (λ_1_–λ_2_), can be used to understand how the formulations recover after stress application as e.g., after extrusion. A difference close to zero means a great ability to recover after stress application, whereas larger values indicate permanent deformation. From the obtained data, WPI emulsion gels were able to fully recover, whereas especially MWP emulsion gels showed a higher level of permanent deformation.

By creep–recovery analysis, it was shown that gels characterized as highly elastic, such as 10% WPI emulsion gels, were not necessarily good food inks for 3D printing, as the extrusion process required weaker materials that could deform and fit through a narrow nozzle. Emulsion gels containing MWP met a narrow range of material properties including a relatively low RS of 68–80% and α between 0.083–0.155. As earlier shown, this led to the continuous extrusion of stable cube structures with smooth layering.

## 4. Conclusions

This study investigated the applicability of heat-induced high protein WPI gels for extrusion-based 3D printing. The effect of ionic strength (0–250 mM NaCl) on printability was assessed, but increasing ionic strength did not result in improved printability of WPI gels. The addition of fat (10% *w/w*) to WPI gel formulations, to form an emulsion gel, increased the gel stiffness and improved printability. However, in order to print a gel in a continuous process, it was necessary to soften the protein network and reduce the elasticity by substituting a fraction of WPI by either MWP or MCI. Especially substitution by MWP resulted in good printability and stable shapes after deposition. Formulations including MCI had tendencies to phase separate and collapse after extrusion. Emulsion gels that had appropriate mechanical strength and viscoelastic properties suitable for 3D printing contained 10% protein and 10% fat and were composed of 8% WPI/2% MWP and 6.5% WPI/3.5% MWP, respectively. These formulations extruded in a continuous form into stable cube structures. Although the printing precision could be improved to increase the appeal to the end user, the results of this study can provide guidance for the formulation design of dairy protein gels used as nutritional food inks for extrusion-based 3D printing.

## Figures and Tables

**Figure 1 foods-10-00008-f001:**
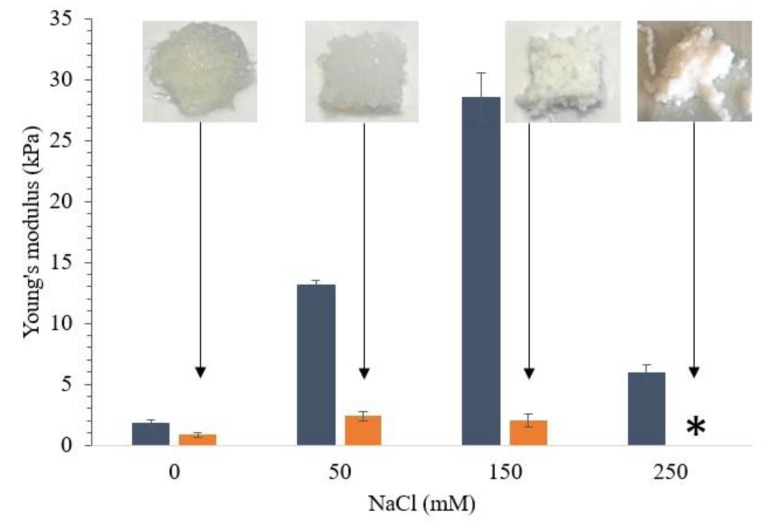
Young’s modulus of gels with 10% protein from whey protein isolate (WPI) with increasing NaCl concentration (0–250 mM). Dark blue bars show Young’s modulus before 3D printing. Orange bars show Young’s modulus after printing. Gels were prepared at pH 7. Samples marked with * could not be measured. Shown are averages of triplicate measurement and error bars as std. deviations.

**Figure 2 foods-10-00008-f002:**
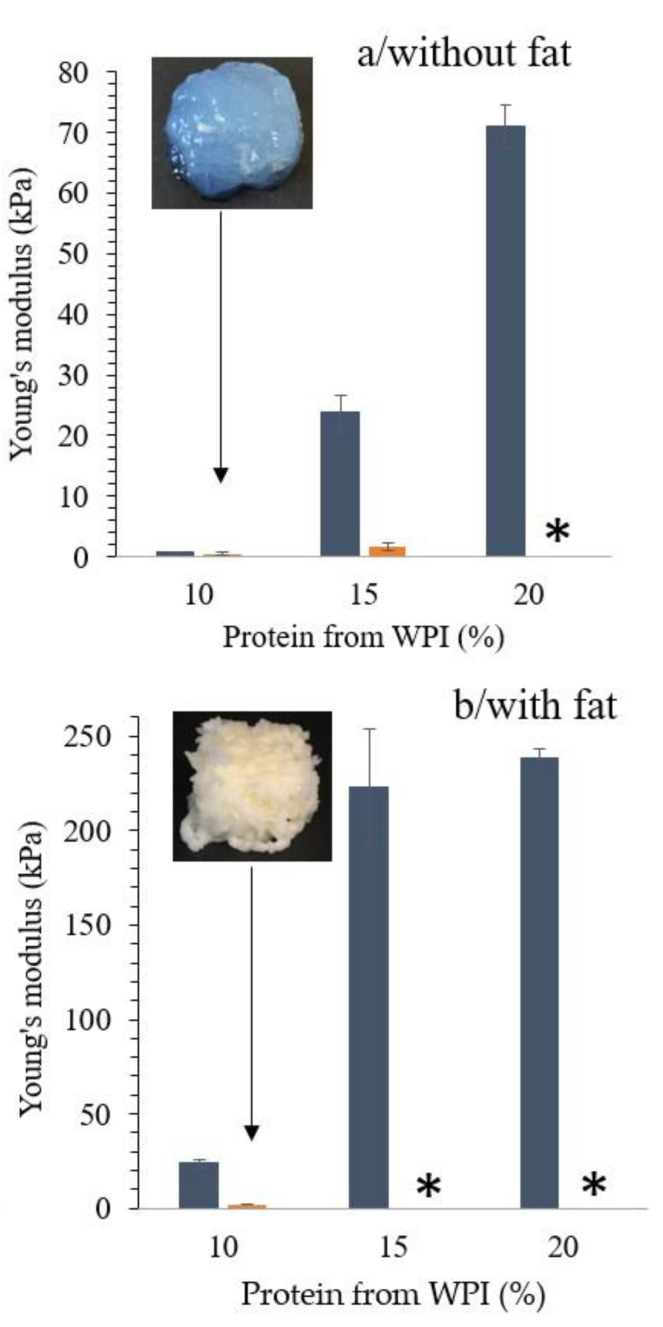
(**a**) Young’s modulus of heat induced WPI gels with 10, 15, and 20% protein before and after 3D printing. (**b**) Young’s modulus of heat-induced WPI emulsion gels with 10, 15, and 20% protein and 10% fat. Dark blue bars present Young’s modulus before printing, orange bars present Young’s modulus after printing. Gels marked with * could not be printed. Shown are averages of triplicate measurement and error bars as std. deviations. Note that the scale of plot (**a**) and (**b**) is different due to significant differences between the samples.

**Figure 3 foods-10-00008-f003:**
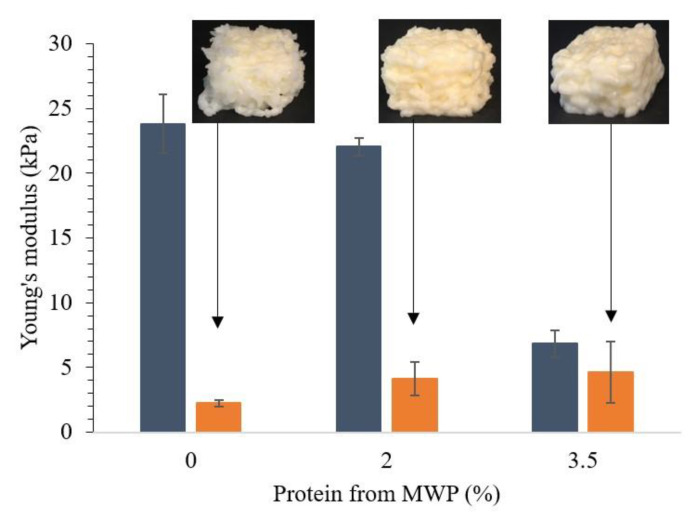
WPI and microparticulated whey protein (MWP) emulsion gels (10% fat *w/w*) with a total protein content of 10%. Emulsion gels formed with 0, 2, and 3.5% protein from MWP and 10, 8, and 6.5% protein from WPI, respectively, before and after 3D printing. Shown are averages of triplicate measurement and error bars as std. deviations.

**Figure 4 foods-10-00008-f004:**
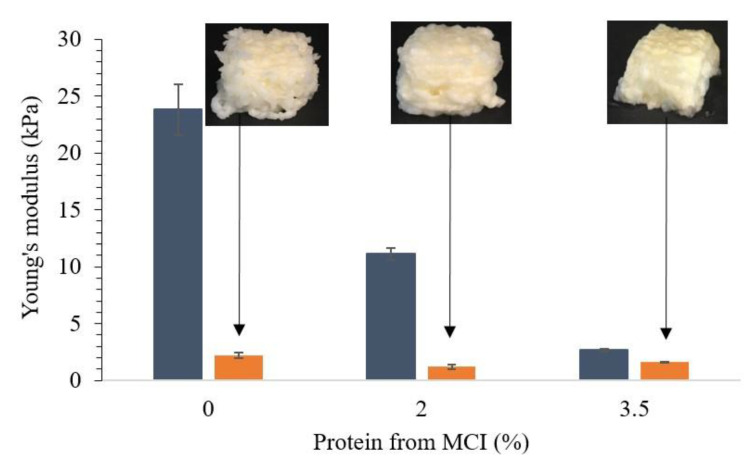
WPI and MCI emulsion gels (10% fat *w/w*) with a total protein content of 10%. Emulsion gels formed with 0, 2, and 3.5% protein from MCI and 10, 8, and 6.5% protein from WPI, respectively, before and after 3D printing. Shown are averages of triplicate measurement and error bars as std. deviations.

**Figure 5 foods-10-00008-f005:**
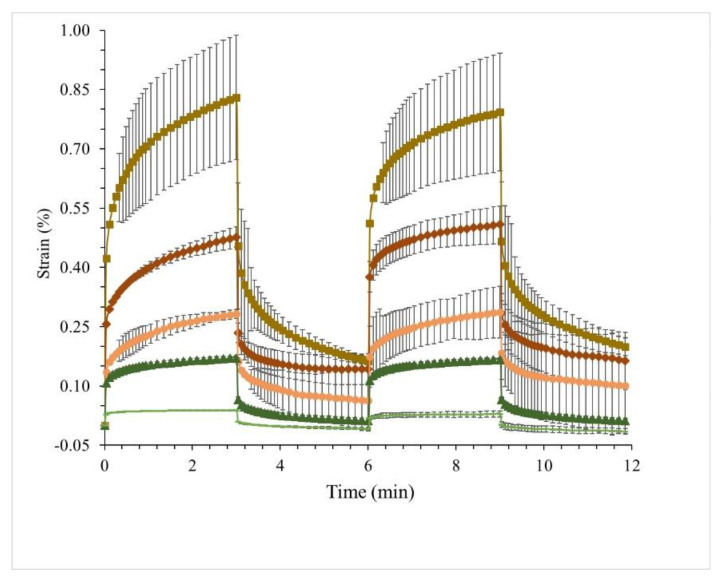
Creep–recovery cycles in two runs of WPI emulsion gels; (■) MWP(3.5), (♦) MWP(2), (●) MCI(3.5), (▲) MCI(2), (▬) WPI(10). Given are means of two repetitions ± std. deviations.

**Table 1 foods-10-00008-t001:** Composition of emulsion gels with 10% protein and 10% fat.

	Protein Source		
Sample Code	WPI (%)	MWP (%)	MCI (%)
WPI(10)	10	0	0
MWP(2)	8	2	0
MWP(3.5)	6.5	3.5	0
MWP(5)	5	5	0
MCI(2)	8	0	2
MCI(3.5)	6.5	0	3.5
MCI(5)	5	0	5

**Table 2 foods-10-00008-t002:** Oscillatory stress sweep measurements: yield stress, storage modulus G’, and loss modulus G’’ of different dairy protein gels.

	Yield Stress (Pa)	Storage Modulus, G’ (Pa)	Loss Modulus, G’’ (Pa)
WPI Gels without Fat			
WPI(10)Fat(0)	139 ± 16	106 ± 12	29 ± 2
WPI(15)Fat(0)	*	12,404 ± 135	1480 ± 30
WPI(20)Fat(0)	*	58,707 ± 1829	8079 ± 67
Emulsion Gels with 10% Fat			
WPI(10)	826 ± 106	2734 ± 711	399 ± 94
WPI(15)	*	41,001 ± 858	5633 ± 195
WPI(20)	*	82,886 ± 698	11,503 ± 39
MWP(2)	382 ± 100	414 ± 42	61 ± 2
MWP(3.5)	525 ± 70	201 ± 37	37 ± 2
MCI(2)	275 ± 9	747 ± 53	120 ± 18
MCI(3.5)	293 ± 58	215 ± 38	46 ± 11

* indicates gels with storage modulus G’ greater than loss modulus G’’ within stresses of 0.1–1000 Pa (no crossover found). Storage modulus G’ and loss modulus G’’ obtained in the linear viscoelastic region (0.1–1 Pa). Given are averages of triplicate measurements ± std. deviations.

**Table 3 foods-10-00008-t003:** % Recovery strain (RS) and fitting parameters for creep–recovery modeling of dairy protein emulsion gels (α—derivative order; λ_1_σ_0_ —creep constant; λ_2_σ_0_ —recovery constant), for applied stress: 1 Pa. Given are parameter averages of the least square modeling of two creep–recovery cycles ± std. deviations.

	Creep 1					Creep 2				
		Parameters of the Model					Parameters of the Model			
	RS%	α	λ_1_σ_0_	λ_2_σ_0_	λ_1_–λ_2_	RS%	α	λ_1_σ_0_	λ_2_σ_0_	λ_1_–λ_2_
WPI(10)	100.0	0.099 ± 0.009	0.035 ± 0.001	0.042 ± 0.002	−0.008	100.0	0.112 ± 0.000	0.025 ± 0.007	0.039 ± 0.001	−0.014
MWP(2)	70.2	0.134 ± 0.010	0.376 ± 0.013	0.307 ± 0.010	0.069	67.8	0.083 ± 0.006	0.451 ± 0.041	0.319 ± 0.022	0.132
MWP(3.5)	80.5	0.155 ± 0.004	0.660 ± 0.137	0.595 ± 0.141	0.065	74.8	0.116 ± 0.010	0.673 ± 0.121	0.531 ± 0.120	0.135
MCI(2)	92.9	0.108 ± 0.008	0.144 ± 0.019	0.144 ± 0.000	0.001	92.9	0.100 ± 0.016	0.145 ± 0.043	0.141 ± 0.008	0.004
MCI(3.5)	77.8	0.185 ± 0.029	0.214 ± 0.020	0.194 ± 0.069	0.020	64.6	0.134 ± 0.026	0.234 ± 0.058	0.167 ± 0.049	0.067

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
