# Peer review of "Formulation of Heat-Induced Whey Protein Gels for Extrusion-Based 3D Printing"

_foods, 2020, doi:10.3390/foods10010008_

Round 1
Reviewer 1 Report
In this manuscript found for a little mistakes:
in line 49 the punctuation mark should be brought closer to the brackets
in lines 54 and 75 should be insert the space between words milk and (Aspri and negligible and (Cunsolo
in lines 79-80 the order of the references provided should be changed by year
in line 81 why the unit of measurement Kg is capitalized and different everywhere else in the text
in line 82 the order of the references provided should be changed by alphabetic
These few errors do not affect the substantive value of the text but require improvement.
I accept after minor revision (corrections to minor errors and text editing).
Author Response
We thank the review for detailed review. We have revised the text editing accordingly.
Reviewer 2 Report
This article is very interesting and the scientific method is well described. Here are a few comments that would help to further improve its quality.
L44-46: The introduction could be enriched by recent articles whose subject is very close to the present study.
Ex. : LEE, C. P., KARYAPPA, R. & HASHIMOTO, M. 2020. 3D printing of milk-based product. RSC Advances, 10, 29821-29828.
L64-66: Data on the typical use and properties of MWP and MCI are missing. Please clarify whether the changes in gel hardness are attributed to changes in ionic strength as indicated L63.
L88: 10%: How was this value chosen?
Table 1: The protein level in the study is set at 10%. This seems low compared to what is stated in the introduction, especially for the enrichment of the diet of the elderly. Is this limitation only technical?
L106: Please specify the method of lubrication and the product used.
L113: Hot gel formation and then cooling. How can the flow be correct at room temperature? Do you have an idea of the pressure in the cartridge?
L117: Are these the pictures in Figures 2, 3 and 4?
L118: Please clarify what the term "quantification" means in this context.
L137-141: Why the 3D printed objects do not have the same shape as those mentioned L114? The difference in surface area is 20 mm² and the difference in height is 10mm (factor 2), please justify this choice.
L174: Can you specify the value of the isoelectric point?
L174-175: The addition of a reference would be appreciated.
L200: The Young's modulus remains low but, however, more than 3 times higher than the announced value L177.
L231-236: Lipids have often been considered, in the literature, to act as lubricants and therefore beneficial to the printability of foodstuffs. What is the opinion of the authors?
L260-263: This passage deserves to be in the "introduction" section.
L371-383: According to the supplied elements, there's no printing with a precise geometry. Can you indicate perspectives to remedy this? Even if this study is intended as an aid to formulation.
L431: Check the year of publication of reference 15.
Author Response
The authors would like to thank the author for a constructive review of our work and suggestions to improve the manuscript.
L44-46: The introduction could be enriched by recent articles whose subject is very close to the present study.
Ex. : LEE, C. P., KARYAPPA, R. & HASHIMOTO, M. 2020. 3D printing of milk-based product. RSC Advances, 10, 29821-29828.
We agree that this reference is relevant and we have added updated the introduction with new references.
L64-66: Data on the typical use and properties of MWP and MCI are missing. Please clarify whether the changes in gel hardness are attributed to changes in ionic strength as indicated L63.
The information missing has been added. The change is not due to changes in ionic strength but to weakening of the gel structure due to steric hindrance effects or thermodynamic incompatibility between the two protein systems leading to phase separation.
L88: 10%: How was this value chosen?
The 10% was based on previous trials in our lab. In future work it will be relevant to test the effect of fat concentration specially for gels with more than 10% protein.
Table 1: The protein level in the study is set at 10%. This seems low compared to what is stated in the introduction, especially for the enrichment of the diet of the elderly. Is this limitation only technical?
Not really, we have decided to start with 10% as these gels had a lower young modulus. For high protein concentration, we should consider to decrease the fat content and this study is ongoing in our lab.
L106: Please specify the method of lubrication and the product used.
The cartridges were lubricated with rapeseed oil and the information was added to the text.
L113: Hot gel formation and then cooling. How can the flow be correct at room temperature? Do you have an idea of the pressure in the cartridge?
This is a good point. The gels were already formed in the cartridges and just extruded at room temperature. We do not have the information about the pressure in the cartridge but agree that it would make sense to obtain it and perform more fundamental studies in this way.
L117: Are these the pictures in Figures 2, 3 and 4?
Yes, it is correct. Originally, we planned to have the pictures as a separated figure but found out that it would be more informative to have them associated to each picture.
L118: Please clarify what the term "quantification" means in this context.
Thank you for noticing this. We meant that we did not quantify the quality of the printability. The sentence was corrected.
L137-141: Why the 3D printed objects do not have the same shape as those mentioned L114? The difference in surface area is 20 mm² and the difference in height is 10mm (factor 2), please justify this choice.
These measurements in these lines refer to the gels before printing. So the gels were removed from the cartridge and cut in cylinders for further analysis.
L174: Can you specify the value of the isoelectric point?
Thank you, the isoelectric point has been added to the manuscript.
L174-175: The addition of a reference would be appreciated.
A reference has been added.
L200: The Young's modulus remains low but, however, more than 3 times higher than the announced value L177.
Good point. The sentence has been changed.
L231-236: Lipids have often been considered, in the literature, to act as lubricants and therefore beneficial to the printability of foodstuffs. What is the opinion of the authors?
We agree with the review, the fat can act as a lubricant as previously mentioned, however in this sentence we refer to the fat entrapped in the emulsion structure.
L260-263: This passage deserves to be in the "introduction" section.
We understand the reviewer’s suggestion. We found out that this information in the introduction made our message less clear towards the objective. Other specifications may also work in this context. For that reason, we would prefer to have it in this section.
L371-383: According to the supplied elements, there's no printing with a precise geometry. Can you indicate perspectives to remedy this? Even if this study is intended as an aid to formulation.
We agree that further work will be needed and a sentence was added to conclusions clarify this matter
L431: Check the year of publication of reference 15.
Thank you, it was corrected